# Herbal Approaches to Pediatric Functional Abdominal Pain

**DOI:** 10.3390/children9081266

**Published:** 2022-08-22

**Authors:** Rebecca N. Cherry, Samra S. Blanchard, Ashish Chogle, Neha R. Santucci, Khyati Mehta, Alexandra C. Russell

**Affiliations:** 1Deep Well Health Care, Elkins Park, PA 19027, USA; 2Division of Pediatric Gastroenterology, University of Maryland School of Medicine, Baltimore, MD 21201, USA; 3Department of Pediatric Gastroenterology, CHOC Children’s, Orange, CA 92868, USA; 4Division of Pediatric Gastroenterology, Hepatology and Nutrition, Cincinnati Children’s Hospital Medical Center, Cincinnati, OH 45229, USA; 5Department of Pediatrics, University of Cincinnati College of Medicine, Cincinnati, OH 45267, USA; 6Division of Pediatric Gastroenterology, Hepatology and Nutrition, Loma Linda University Medical Center, Loma Linda, CA 92354, USA; 7Division of Pediatric Gastroenterology, Department of Pediatrics, Vanderbilt University Medical Center, Nashville, TN 37232, USA

**Keywords:** abdominal pain

## Abstract

Chronic abdominal pain is one of the most common problems seen by both pediatricians and pediatric gastroenterologists. Abdominal-pain-related functional gastrointestinal disorders (AP-FGIDs) are diagnosed in children with chronic and recurrent abdominal pain meeting clinical criteria set forth in the Rome IV criteria. AP-FGIDs affect approximately 20% of children worldwide and include functional dyspepsia (FD), irritable bowel syndrome (IBS), functional abdominal pain (FAP), and abdominal migraine. IBS accounts for 45% of pediatric AP-FGIDs. The pathophysiology of functional abdominal pain involves an interplay of factors including early life events, genetics, psychosocial influences, and physiologic factors of visceral sensitivity, motility disturbance, altered mucosal immune function, and altered central nervous system processing. Treatment approaches are varied and can include dietary, pharmacologic, and complementary medicine interventions, as well as psychosocial support, depending on the many aspects of the disorder and the needs of the individual patient. There is a strong interest in complementary and integrative medicine approaches to pediatric pain from both patients, providers, and families. In this article, we discuss popular herbal treatments typically used in the field of complementary medicine to treat pediatric AP-FGIDs: peppermint oil, Iberogast^®^, cannabis, fennel, and licorice. While high-quality data are rather limited, studies generally show that these remedies are at least as effective as placebo, and are well tolerated with minimal side effects. We will need more placebo-controlled, double-blind, and unbiased prospective studies to document and quantify efficacy.

## 1. Introduction

Chronic abdominal pain is one of the most common gastrointestinal symptoms seen by both pediatricians and pediatric gastroenterologists. Abdominal-pain-related functional gastrointestinal disorders (AP-FGIDs) are diagnosed in children with chronic and recurrent abdominal pain meeting clinical criteria set forth in the Rome IV criteria [1]. AP-FGIDs affect approximately 20% of children around the world and include patients with functional abdominal pain (FAP), functional dyspepsia (FD), abdominal migraine, and irritable bowel syndrome (IBS) [2,3]. IBS accounts for 45% of pediatric AP-FGIDs [2].

The pathophysiology of functional abdominal pain involves an interplay of multiple factors including early life experiences, genetics, psychosocial influences, and physiologic factors of visceral sensitivity with altered central nervous system processing, abnormal motility, and abnormal mucosal immunity [3]. Treatment approaches are varied and can include dietary, pharmacologic, and complementary medicine interventions, as well as psychosocial support, depending on the many aspects of the disorder and the needs of the individual patient. In light of the strong interest in complementary and integrative approaches to pediatric pain among patients, providers, and families [4], this paper describes some of the most popular herbs for treatment of abdominal pain in children. We do not attempt to lay out a detailed exposition of the studies which have already been done, but rather offer a general overview of some familiar herbal treatments without advocating for or against their use.

## 2. Methods

In the following sections, we present an overview consisting of a non-systematic summation and analysis of the literature available on the topic of herbal approaches to pediatric functional abdominal pain, noting that there is a paucity of data in this area. The herbs selected for review in this paper were determined by consensus among pediatric gastroenterologists with expertise in Integrative Medicine who also participate in the Integrative Medicine Special Interest Group of the North American Society of Pediatric Gastroenterology, Hepatology, and Nutrition (NASPGHAN). We selected six herbal treatments that are commonly used in clinical practice and which have been addressed in the medical literature. Relevant articles were identified using the PubMed, Cochrane, and Scopus databases, searched between January and May 2022. Search terms included “peppermint,” “Mentha × peperita,” “fennel,” “Foeniculum vulgare,” “licorice,” “Glycyrrhiza glabra,” ”Iberogast,” “STW5,” “marijuana,” “cannabis,” “cannabidiol,” “THC,” “tetrahydrocannabinol,” “ginger,” “Zingiber,” and “herbal medications” in combination with the terms “clinical trial,” “review,” “systematic review,” “meta-analysis,” and “pediatric” or “children.” We included full manuscripts, prospective studies, retrospective studies, review articles, systematic reviews, and meta-analyses. We excluded studies that were not published in the English language. We did not perform meta-analyses or standardized literature reviews, but rather have presented highlights and impressions which may be of use to fellow clinicians. These are also summarized in Table 1, “Summary of clinically relevant findings”.

## 3. Peppermint Oil

Peppermint oil (PMO) is extracted from fresh peppermint leaves by steam distillation and contains menthol as the primary ingredient [13]. PMO has a therapeutic effect in patients with functional gastrointestinal disorders due to its actions at various levels of the microbiome–gut–brain axis. It exerts a spasmolytic effect on smooth muscle cells by decreasing calcium influx via inhibition of L-type Ca^2+^ channels and reduction of acetylcholine release from enteric nerves via its action on nicotinic receptors in the GI tract [14,15]. The use of PMO for pediatric FGID patients has been examined in several studies. A randomized, double-blind controlled trial by Kline et al. examined the efficacy of pH-dependent, enteric-coated, PMO capsules in the treatment of IBS symptoms in 50 children (8–17 years of age). Children with IBS were given either pH-dependent, enteric-coated PMO capsules (Colpermin^®^—187 mg of peppermint oil) or placebo (Arachis oil capsule) for 2 weeks. Patients weighing 45 kg or more received two PMO or placebo capsules three times a day. Children with weights between 30 kg to 45 kg were given one capsule three times a day. Of the 42 children that completed the study, 75% who received PMO reported significant improvement in severity of abdominal pain as compared to 19% in the placebo group (*p* < 0.001) [14]. Similar results were also seen in a randomized controlled study performed by Asgarshirazi et al. that included 120 children (4–13 years of age) with functional abdominal pain, functional abdominal pain syndrome, functional dyspepsia, and IBS [6]. Children were randomized to receive either Lactol^®^ capsules (Bacillus coagulans + fructooligosaccharide), PMO (Colpermin^®^—187 mg of peppermint oil), or a placebo (folic acid) for one month. The dosing protocol in this study was the same as mentioned in the Kline study above. From the data analysis of 88 children who completed the study, significant improvements in pain duration, severity, and frequency were seen in the PMO group as compared to placebo (*p* = 0.0001, *p* = 0.0001, and *p* = 0.001, respectively). PMO was also found to be superior to Lactol in decreasing pain duration and severity (*p* = 0.040 and *p* = 0.013, respectively). No side effects were reported in any of the study groups.

There has been one pediatric study to date assessing the motility effects of PMO in children. This study included 30 children with functional abdominal pain and demonstrated a decrease in whole gut mean peak contraction with PMO but without any significant change in the whole gut transit time [16]. This could explain the effectiveness of PMO in relieving pain in IBS without having a significant impact on the gastrointestinal transit time.

Peppermint oil is generally well tolerated at the commonly recommended dosage. It is contraindicated in patients with hiatal hernia or gastroesophageal reflux disease (GERD) as it can worsen reflux symptoms due to its effects on the lower esophageal sphincter [17,18].

## 4. Fennel

Fennel, also known as Foeniculum vulgare, is an herb that grows year long and is one of the oldest plants utilized in traditional medicine [19]. It possesses anti-inflammatory, antioxidant, antispasmodic, and carminative properties [20,21]. Anethole, a component of fennel oil, improves gastric emptying and gastric accommodation in rats, suggesting potential beneficial effects in FD [22]. Given these multiple properties, fennel has been used in different forms over the years for various gastrointestinal symptoms.

Pediatric studies assessing the efficacy of fennel for FGIDs are lacking, although fennel tea has been explored for the treatment of infant colic. In a descriptive study assessing the usage of herbal preparations in Turkey, fennel tea was used by approximately 77% of mothers to treat their children’s gas pain and constipation [23]. A systematic review on herbs demonstrated the efficacy of fennel in different preparations (seed oil, tea, or in combination with other herbs such as Colimil^®^) to reduce episodes of crying in infants with colic (*n* = 461, age range 2–12 weeks old) through four randomized controlled trials (RCT) [7]. No side effects were reported in these studies [8,9,10,11].

Studies in adults with IBS have shown some promising findings. In an RCT of 121 adults with IBS, fennel essential oil combined with curcumin improved IBS symptom severity scores and quality of life. A higher response was noted in the treatment compared with the control group (25.9% vs. 6.8%, *p* = 0.005). The mixture was found to be safe and well-tolerated [21]. Similarly, an open-label study demonstrated the efficacy of Enterofytol^®^, a product with extracts of turmeric and fennel oil, in improving symptom severity index and quality of life in 211 IBS patients, especially those with diarrhea-predominant IBS [24]. A double-blinded controlled study from Germany evaluated the efficacy of Lomatol^®^ drops (combination of caraway fruit extracts, fruits of fennel, peppermint leaves, and wormwood) in comparison with metoclopramide drops in patients ages 18–85 years with upper abdominal symptoms (*n* = 60). Lomatol^®^ drops were superior to metoclopramide with greater improvement in abdominal pain (*p* = 0.02), nausea (*p* = 0.02), heartburn (*p* = 0.016), and gastric spasms (*p* = 0.002). It was also better tolerated with fewer adverse effects than treatment with metoclopramide [25]. While these data are favorable, pediatric studies are necessary to establish safety and efficacy data in children.

## 5. Licorice

Licorice root (Glycyrrhiza glabra) may have promise for the treatment of pediatric FGIDs. The main constituent of licorice root is the triterpenoid saponin glycyrrhizin (also known as glycyrrhizic acid or glycyrrhizinic acid), which is usually found in concentrations ranging from 6% to 10%. The intestinal flora is believed to hydrolyze glycyrrhizin, yielding the aglycone molecule (glycyrrhetinic acid) and a sugar moiety, which are both absorbed [26,27]. The active components of DGL are flavonoids. Other active constituents of licorice include isoflavonoids (e.g., isoflavonol, kumatakenin, licoricone, glabrol); chalcones; coumarins (e.g., umbelliferone, herniarin); triterpenoids; and sterols, lignins, amino acids, amines, gums, and volatile oils [27].

Deglycyrrhizinated licorice (DGL), a processed licorice extract made by removing the glycyrrhizin molecule to eliminate its mineralocorticoid properties, has been used for several gastrointestinal disorders secondary to its anti-inflammatory, antibacterial, and antiviral pharmacologic actions [27]. DGL is also used in the setting of gastric hyperacidity and functional dyspepsia. However, rather than inhibiting acid release, DGL stimulates the standard defense mechanisms that prevent ulcer formation and promote mucosal healing by increasing blood supply and mucus production [28,29]. In a small, randomized, double-blind, placebo-controlled study of 50 adults with functional dyspepsia as diagnosed by Rome III criteria, subjects were randomized to placebo or a 75 mg extract of Glycyrrhiza glabra (GutGard^®^, Karnataka, India) for 30 days. Compared to placebo, those receiving the licorice extract showed a significant decrease in total symptom scores and improved quality of life [30].

The dose of licorice for clinical use is based on the content of glycyrrhetinic acid. The standard dosage for DGL in adults is two to four 380 mg chewable tablets 20 min before meals [31]. It is essential to review with patients that DGL is a safer formulation than those containing glycyrrhizin as glycyrrhizin has mineralocorticoid properties that can cause hypertension and hypokalemia-induced secondary disorders [32,33]. There are currently no pediatric studies confirming safety, dosing, or benefit for children with FGIDs.

## 6. STW5

STW5 or Iberogast^®^ is a combination of multiple herbs including an extract from bitter candytuft (Iberis amara), angelica root (Angelica radix), milk thistle (Silybi mariani fructus), celandine herb (Chelidonium majus), caraway fruit (carvi fructus), licorice root (Liquiritiae radix), peppermint (Menthae piperitae folium), lemon balm leaves (Melissa folium), and chamomile flower (Matricariae flos). Bitter candytuft (I. amara) selectively inhibits binding to muscarinic M3 receptors, while extracts of celandine and chamomile inhibit binding to 5-HT4 receptors and licorice root to 5-HT3 receptors [34,35]. (Peppermint’s effects on the GI tract are discussed in more detail in Section 2, “Peppermint Oil,” and licorice is discussed in Section 4, “Licorice.”) However, the relative importance of each component’s mechanism of action to the overall effects of STW5 use is uncertain.

STW5 has been demonstrated to have multiple effects on the gastrointestinal tract that are conducive to improvement in FGID symptoms. These include modulating visceral hypersensitivity, gastric motility, and gastric accommodation [36]. In rats, oral administration of STW5 showed reduced jejunal afferent sensitivity to mechanical distension, serotonin, and bradykinin [37]. The improvement seen in IBS and functional dyspepsia patients may be attributable to the influence of STW5 on visceral hypersensitivity. STW5 also has organ-specific effects on the gastric fundus and antrum. STW5 has a significant, dose-dependent muscle relaxant effect in the fundus, which can help accommodation in functional dyspepsia, while STW5 stimulates phasic contractility in the antrum [38,39,40].

STW5 has some pediatric clinical data to support its use. A study of 980 children who received STW5, 10–20 drops three times a day for 7 days found improvement in upper and lower gastrointestinal symptoms. Thirty-nine percent of children reported complete relief of their symptoms. STW5 was well tolerated in 94.8% of cases with only four mild adverse events [12].

STW5 has been extensively evaluated for tolerability and side effects since it was introduced to the market in the 1960s. In three large studies, few patients developed abdominal pain, pruritus, alopecia, hypersensitivity, hypertension, vomiting, and headaches as adverse effects but this was not statistically significant compared to placebo [12,41,42]. The first case of acute liver failure was reported in 2018 in Germany and has subsequently been described in Denmark and Switzerland [43,44,45]. The WHO database (VigiBase) contains 70 cases of hepatobiliary disorders associated with STW5 [46]. Among nine herbs in STW5, only Chelidonium majus (Greater celandine) has been associated with liver toxicity. No liver toxicity was reported with other extracts [47]. To develop hepatic side effects, the doses are usually 100–200 times higher than the amount in STW5 [48]. The pathophysiology responsible for the reported herb-induced liver injury (HILI) is mostly compatible with idiosyncratic reaction.

Iberogast^®^ comes as liquid drops. The listed dose for children from 3 to 5 years is ten drops three times a day, for children 6 to 12 years 15 drops three times a day, and for adolescents and adults 20 drops three times a day. It is taken with a small amount of liquid before or during meals.

## 7. Cannabis

Marijuana, or cannabis, is known for both medicinal and recreational uses. The Cannabis sativa plant produces a range of compounds, primary among which are the cannabinoids. The major cannabinoids, delta-9-tetrahydrocannabinol (THC) and cannabidiol (CBD) are produced in large amounts by cultivated cannabis plants and used for their psychoactive and therapeutic effects. Over 100 minor cannabinoids are present in smaller amounts. Cannabinoids exert their effects through the endocannabinoid system, with the primary receptors being CB1 and CB2. Additional receptors, including GPR55, GPR119, PPAR-alpha, PPAR-gamma, and TRPV1, may respond to cannabinoids and modulate CB receptor activity [49].

THC is perhaps best known for its psychoactive effects, but also is used by many to stimulate appetite, decrease nausea, and reduce pain and anxiety. Medical marijuana is not universally legal in the United States. However, a synthetic form of THC, dronabinol, is FDA-approved in the adult population for appetite stimulation in the setting of AIDS as well as for nausea and vomiting in the setting of cancer chemotherapy. There is very little research on the use of THC for treatment of functional pain syndromes. As of this writing, according to clinicaltrials.gov, there are no controlled trials of THC or dronabinol in children for treatment of GI symptoms or abdominal pain. Of note, THC use may have the unintended effect of causing severe recurrent vomiting, a condition known as cannabinoid hyperemesis syndrome (CH). Patients may continue to escalate their cannabis use after developing CH in an attempt to manage their nausea and vomiting, leading to a persistent cycle of symptoms.

In contrast to THC, CBD does not have psychoactive effects. It has been or is currently being studied in children and adolescents for indications including spasticity, epilepsy, and behavioral disorders in the setting of autism or intellectual disability. However, there are no controlled trials of CBD in children for the purposes of treating GI symptoms or abdominal pain.

Among adults, one study has investigated the effectiveness of CBD for functional abdominal pain. A placebo-controlled trial of 50 women with IBS, using chewing gum formulated with 50 mg CBD, found no statistically significant differences in pain scores among those in the CBD group vs. placebo [50]. Adolescent IBD patients using CBD oil have reported in surveys that they experience improved sleep quality, decreased nausea, and better appetite [51].

There are remarkably little data about the use of cannabis for management of abdominal pain, particularly in the setting of functional disorders rather than in inflammatory bowel disease. Enthusiasm over the use of cannabis as a natural product for treatment of abdominal pain must be tempered both by the association of regular use with cannabis hyperemesis syndrome and the impaired executive functioning and school performance associated with adolescent cannabis or THC use.

## 8. Ginger

The rhizome of the ginger plant, Zingiber officinale, has traditionally been used both as a dietary flavoring and as an herbal medication, with its first documentation in Chinese medicine dating to 400 BC. Chemically, ginger contains over 400 compounds, prominent among which are phenols including gingerol and shogaol, and terpenes including zingiberene and bisabolene [52].

Physiologically, ginger has been found to accelerate gastric emptying by 24% at a dose of 1.2 g powdered ginger root in capsule form in a double-blind study of 11 adults with functional dyspepsia, although this change was not associated with an improvement in GI symptoms as measured on a visual analogue scale. The authors did not identify any changes in levels of motilin, ghrelin, or GLP-1 which might explain the increase in emptying rate, although note that they did not check cholecystokinin levels [53]. A study of 13 young adults with a history of motion sickness found that 1 g or 2 g powdered ginger root in capsule form, given 1 h before being spun in a seated position, decreased nausea scores by 30%, prolonged the time to nausea onset by up to 52%, and decreased recovery time by up to 25%. In a subset of this group studied with electrogastrography, ginger pretreatment decreased tachygastric activity by up to 29% [54].

The effects of ginger on chemotherapy-induced nausea and vomiting and for nausea and vomiting during pregnancy have been described in multiple studies, with a recent systematic review concluding that a divided daily dosage of 1500 mg ginger is beneficial for relief of nausea [52]. There are limited data on ginger as an antiemetic in children, however. These studies are described below. Similarly, there are a few studies investigating the use of ginger for functional gastrointestinal disorders including pain and bloating in adults [55,56], but data on the efficacy of ginger for abdominal pain in children are unavailable.

A randomized controlled study of children and young adults aged 8–21 years receiving cisplatin/doxorubicin for osteosarcoma or malignant fibrohistiocytoma compared the effects of powdered ginger and placebo. A total of 60 chemotherapy cycles were studied. Patients weighing 20–40 kg received 1 g of ginger powder or a placebo powder divided into three daily doses on the three days when they received chemotherapy, and patients weighing 40–60 kg received 2 g in three daily doses for the three days. There was a decreased rate of acute moderate to severe nausea (55.6% vs. 93.3%, *p* = 0.003), acute moderate to severe vomiting (33.3% vs. 76.7%, *p* = 0.002), delayed moderate to severe nausea (25.9% vs. 73.3%, *p* < 0.001) and delayed moderate to severe vomiting (14.8% vs. 46.7%, *p* = 0.022) in the ginger group compared to the placebo group [57].

A study of 150 children aged 1–10 years with vomiting due to acute gastroenteritis compared the effects of ginger vs. placebo. A total of 33% of patients treated with ginger had resolution of vomiting after the first dose, compared to 13% in the placebo arm (*p* = 0.003). The ginger product used was a liquid containing 10 mg of ginger extract per 20 drops (1% ginger by weight), given at a dose of 10 mg every 8 h until the resolution of vomiting [58].

Ginger is considered safe in the amounts typically found in foods, with few adverse effects reported in clinical trials. Doses above 5 g per day are more likely to cause side effects. The most common adverse effects are irritation of the mouth and throat, abdominal discomfort, heartburn, burping, and diarrhea. Some of these symptoms may be avoided or reduced by taking ginger in capsule form. Constipation has also been reported with ginger. Ginger bolus from insufficiently chewed pieces of raw ginger, leading to small bowel obstruction, has been reported in four cases [59].

## 9. Conclusions

There is a range of herbal medicines which may improve symptoms in pediatric patients suffering from pain-related FGIDs. However, there are so far very few pediatric studies to confirm their efficacy, dosage, safety, and tolerability, with most data available currently regarding the use of peppermint oil. There are more data available on the use of herbal compounds in adults, but many of these trials have been small, observational, use preparations or formulations which are not commercially available, or are not yet replicated to confirm their findings. Thus, limitations include the paucity of clinical studies available and an absence of reporting on the strength of the evidence provided in each study.

Herbs have the potential to be additional tools for pediatric providers to have in their arsenal, as many are generally well-tolerated and inexpensive. Additionally, patients and families are interested in complementary medicine approaches for symptom management. Many patients perceive herbal treatments as more natural and thus more benign, without an awareness that these same treatments can cause untoward side effects, drug reactions, and interactions. These patients may prefer herbal treatments to manufactured pharmaceuticals and are appreciative of recommendations and guidance from their health care team [4]. The confidence that families have in herbal supplements, in conjunction with recommendations from a trusted physician, may also enhance the placebo effect, boosting the effectiveness of herbal treatments in “real world” use [60]. Further research is necessary for health care providers to offer a holistic medical approach while also providing evidence-based, safe, and cost-effective care.

## Figures and Tables

**Table 1 children-09-01266-t001:** Summary of clinically relevant findings.

	Herbal Compound	Summary of Clinically Relevant Findings
1	Peppermint oil	Efficacy in functional abdominal pain syndromes supported by two placebo-controlled pediatric trials [5,6], using a dose of 187 mg three times daily (for children between 30 and 45 kg) or 374 mg three times daily (for children ≥45 kg). Commercial formulations are readily available. Use is associated with increased symptoms of gastroesophageal reflux. Enteric-release formulations used in the published studies are not suitable for children unable to swallow pills.
2	Fennel	Fennel has been shown to reduce episodes of crying in infants with colic through four randomized controlled trials (RCT) [7] No safety signals were identified in these studies [8,9,10,11]. Dosage and formulation used in the controlled trials were inconsistent. Pediatric studies assessing the efficacy of fennel for FGIDs are lacking.
3	Licorice	There are currently no pediatric studies confirming safety, dosing, or benefit for children with FGIDs. Licorice that has not been deglycyrrhizinated has mineralocorticoid properties that can cause hypertension and hypokalemia.
4	STW5	In a large pediatric study, 39% of children who received STW5, 10–20 drops three times a day for 7 days reported complete relief of upper and lower GI symptoms. STW5 was well tolerated in 94.8% of cases with only four mild adverse events [12]. Dosing is uncomplicated: for ages 3 to 5 years: 10 drops three times a day; for ages 6 to 12 years: 15 drops three times a day; for teens and adults, 20 drops three times a day. Liquid formulation is considered by some to be unpalatable.
5	Cannabis	There are no pediatric data on cannabis for functional gastrointestinal disorders. Tetrahydrocannabinol (THC), found in many cannabis products, has significant risk of side effects including long-term developmental problems.
6	Ginger	There are no data on ginger for functional abdominal pain in children, although it has been studied for nausea and vomiting. Ginger formulations are not standardized.

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
