# Peer review of "Herbal Approaches to Pediatric Functional Abdominal Pain"

_children, 2022, doi:10.3390/children9081266_

Round 1
Reviewer 1 Report
The manuscript is an interesting piece refecting the popular interest about herbal medicine, but unfortunately it lacks the clear demonstration of the benefits of that approach. Clearly there is a bias towards the presentation of positive findings "Herbal medicine, also known as phytotherapy, can significantly improve symptoms in pediatric patients suffering from pain related FGIDs. These can be wonderful tools for pediatric providers to have in their arsenal, as they are generally well-tolerated, effective, and inexpensive" while the systematic reviews (as for ex ref 25) are far more prudent in their conclusions. Sensible, scientific sound advice is welcome in this topic but most of the references and type of studies cited are of very low quality or impact factor. Conclusions should be far more moderate.
Author Response
We addressed the statement and presenting the herbal therapy as an alternative approach and changed our text.
Reviewer 2 Report
The manuscript "Herbal approaches to pediatric functional abdominal pain" is a narrative review related to symptomatic treatment with herbal medicine of chronic abdominal pain in pediatric patients. Functional digestive disorders associated with abdominal pain are diverse and the review includes disorders in infants as well as in preschool and school-age children, which introduces a high risk of bias for the authors of the work. The basic information to support the pharmacological effects is insufficient and, in many cases, anecdotal; the clinical studies are scarce and inconclusive, especially because the outcome variables are subjective.
The value of the study is likely to be the information on the various herbal treatments used in children with abdominal pain, as this trend is widely used. As the authors comment, basic and clinical studies are required for each of the alternatives with robust designs that allow their safety and efficacy to be established.
Author Response
We agree that the most pediatric studies are anectodal and we also reviewed the largest pediatric studies which are published. We eliminated any statement introducing bias.
Reviewer 3 Report
This is a comprehensive review on the use of herbal treatment for abdominal pain in children.
Overall, the topics are too extensive and hard to read. I would suggest to shorten the information and stick to the most important implications, which are current available data and from there possible clinical appliance in children.
Also, I would suggest to use kilograms and not pounds (lines 92 and 93).
Author Response
We shorten the text as suggested and used kilograms.
Round 2
Reviewer 1 Report
The current version of the manuscript is more adjusted to the ongoing controversies around the use of herbal medicine.
Still a few points need to be addressed:
1. Peppermint oil: Mention to studies of refs 19 and 20 refer "per protocol" analysis, which is considerably more favourable than "intention to treat". In fact in the study of Asgarshirazi et al (ref 20), frequency and severity of pain improved on placebo and lactol. The authors recommend "Pepper- mint tea can be made brewing one to two teaspoons of dry leaves steeped in 8 oz hot water as needed". Treatments are usually recommended at clearly defined doses while the authors recommend imprecise terms.
2. Mentions to Lomatol and Iberogast are inappropriate as these are brand names.
3. Study of ref 49 of a pediatric study is only available in Abstract form and was never published... data are quite limited, therefore of questionable consistency if not published in the following 11 years.
Author Response
The manuscript is an interesting piece refecting the popular interest about herbal medicine, but unfortunately it lacks the clear demonstration of the benefits of that approach. Clearly there is a bias towards the presentation of positive findings "Herbal medicine, also known as phytotherapy, can significantly improve symptoms in pediatric patients suffering from pain related FGIDs. These can be wonderful tools for pediatric providers to have in their arsenal, as they are generally well-tolerated, effective, and inexpensive" while the systematic reviews (as for ex ref 25) are far more prudent in their conclusions. Sensible, scientific sound advice is welcome in this topic but most of the references and type of studies cited are of very low quality or impact factor. Conclusions should be far more moderate.
Answer:
We changed our conclusion as we agree with reviewer that it is a strong statement.
“Herbal medicine, also known as phytotherapy, may improve symptoms in pediatric patients suffering from pain related FGIDs. While there is a dearth of pediatric and pharmacologic studies on herbal medications, they merit further examination in future studies. Herbs have the potential to be additional tools for pediatric providers to have in their arsenal, as they are generally well-tolerated and inexpensive.”
- Peppermint oil: Mention to studies of refs 19 and 20 refer "per protocol" analysis, which is considerably more favourable than "intention to treat". In fact in the study of Asgarshirazi et al (ref 20), frequency and severity of pain improved on placebo and lactol. The authors recommend "Pepper- mint tea can be made brewing one to two teaspoons of dry leaves steeped in 8 oz hot water as needed". Treatments are usually recommended at clearly defined doses while the authors recommend imprecise terms.
We changed and left as generic information, not specifying the dose and brewing techniques.
- Mentions to Lomatol and Iberogast are inappropriate as these are brand names.
Answer: iberogast is changed to STW5. Lomatol is a combined herbs and we left it.
- Study of ref 49 of a pediatric study is only available in Abstract form and was never published... data are quite limited, therefore of questionable consistency if not published in the following 11 years.
I agree with reviewer that it is only in abstract form. But due to limited studies, we would like to keep it in our manuscript.
Reviewer 2 Report
417 / 5.000
Resultados de traducción
Although the authors have reduced the risk of bias, robust information related to the safety and efficacy of therapeutic proposals is scarce or absent, which prevents them from being included in therapeutic recommendation guidelines. The value of the work is that it informs about treatment alternatives that are probably used as an alternative approach in the management of functional abdominal pain in children.
Author Response
We are recommending as supplementary therapy. We reviewed available sde effects profiles.
We deleted the information regarding infants and also any other gastrointestinal diseases other than functional abdominal pain. We reviewed pediatric studies available in the literature and agree with reviewer that they are anectodal. Our goal is to give information.